# The Relationship between Posture and Muscle Tensive Dysphonia in Teachers: A Systematic Scoping Review

**DOI:** 10.3390/jfmk9020060

**Published:** 2024-03-28

**Authors:** Roberta Franzone, Luca Petrigna, Daniele Signorelli, Giuseppe Musumeci

**Affiliations:** Department of Biomedical and Biotechnological Sciences, Section of Anatomy, Histology and Movement Science, School of Medicine, University of Catania, Via S. Sofia 97, 95123 Catania, Italy; robertafranzone@gmail.com (R.F.); luca.petrigna@unict.it (L.P.); danielesigno92@gmail.com (D.S.)

**Keywords:** posture, taught, muscle tensive dysphonia, voice, MTD

## Abstract

Teachers usually present work-related pain such as neck pain. Their posture could be the cause of these problems; indeed, it is often a sway-back posture. Furthermore, teachers can also experience problems with their voice such as dysphonia, specifically muscle tension dysphonia (MTD). This scoping review aims to find the correlation between teachers’ posture and MTD. It also studies how a posture-based treatment can influence this disorder. Randomized controlled trials, controlled clinical trials, prospective cohort studies, and cross-sectional studies that considered the relationship between posture and MTD and that included teachers in their sample. The search led to an initial number of 396 articles; after the screening process, a final number of eight articles were included. A total of 303 patients were analyzed and all showed altered alignment of the head around the cervical spine with hypertonus of the cricothyroid, suprahyoid, and sternocleidomastoid muscles. Although MTD is a disorder with a multifactorial etiology, the articles revealed a correlation between posture and MTD related to a forward protraction of the cervical spine with a hypertonus of the laryngeal and hyoid musculature. This study also detected that an intervention in posture could reduce vocal disorders.

## 1. Introduction

The voice, along with gestures, is the primary medium through which humans can express thoughts and emotions [1]. To use a simplistic definition, the voice is nothing more than the result of sounds produced by the air passing through the vocal folds in the larynx [2]. In its production, articulation, and modulation, the lower respiratory system, the larynx, and the upper respiratory system are involved. Thus, the voice is not just basic sound; rather, it is a spoken or sung expression [3]. We can distinguish subsets that make language. These are breathing, phonation, articulation, resonance, and prosody (Fuchs). The literature tends to unite the subsets to assess their overall appearance in dysarthria diagnoses. But, on a didactic level, it may be useful to analyze the five subsets separately and only later unite them in a broader overview [4]. Breathing comes into play in the expiratory phase. As the air leaves the lungs, it passes through the two pairs of vocal folds, which, due to the pressure created, adduct to each other, producing vibrations that emit sound, the Bernoulli effect [5]. Phonation occurs when sound is emitted through the excitation of the vocal folds, which vibrate. These vibrations are controlled and modified by the adductor muscles, allowing the creation of what we all know as voice [6,7]. The voice has four main traits. These are extension (its range of sounds), intensity (its power), height (its frequency), and timbre (the most personal part of the voice that lets us tell similar voices apart) [8].

The possibility and ability to master the voice characteristics is a matter for the respiratory muscles and the elastic return of the thorax [9]. Perceiving and controlling these structures lets us control the amount of air and the force with which it passes through the lungs, bronchi, and trachea to reach the larynx. However, only with conscious use of the cervical and laryngeal musculature can the sound be modified in extension and timbre [10]. The tongue, teeth, and nose allow us the articulation of words [10]. Articulation goes hand in hand with resonance, in that a resonant voice is the combination of the adaptation of articulation, tone, and prosody [11]. The latter is the union of syllabic rhythm, word accentuation, and intonation that varies according to language and dialect [12].

Many pathologies and disorders can alter the voice because of the large number of structures involved in voice production. These include vocal folds nodules [13], chord polyps and cysts [14], congenital vocal folds abnormalities [15], vocal folds granulomas [16], oedemas [17], tumor pathologies [15], vocal folds paralysis, pharyngolaryngeal reflux [18], and laryngeal papillomatosis [19]. In addition to the above pathologies, there is one disorder that afflicts many people: dysphonia [20]. Changing behaviors could be useful in the prevention or treatment of voice disorders, and some coaches work on these aspects [21].

Dysphonia is an alteration in voice quality and quantity. It can be classified into organic and functional [20]. The organic can be structural and neurogenic. It is characterized by congenital malformations, inflammation in the larynx, tumors, and trauma. The functional is behavioral, and is characterized by incorrect and/or excessive use of the voice in which the pathophysiological process is not entirely clear. It is thought to be multifactorial. Factors include smoking, gastro-esophageal reflux, stress, respiratory infections, posture, and others [20]. Among the most common functional dysphonia is muscle tensive dysphonia (MTD), the causes of which are still being studied. It is divided into the primary (normal larynx but there is persistent hypertonus of the laryngeal muscles and those surrounding it), and secondary (abnormality of the larynx causing increased use of certain muscles as compensation) [22]. MTD’s symptoms are hoarseness (the need to clear the voice), dryness in the throat, pain, pauses during phonation, and increased difficulty speaking quietly [22]. Given the triggers of functional dysphonia and the traits of MTD, it is easy to imagine how certain people, like singers and teachers, are more affected by this condition. Teachers frequently present voice disorders [23]. It is due to their work, and consequently it is a population that requires attention [24]. Paying more attention to teachers, it is estimated that around 40% of them, aged between 40 and 60, suffer from dysphonia [25,26]. The high percentage can be explained by the fact that teachers are forced to use their voice for a long time and very often in a high tone. The age of the pupils can both cause and exacerbate this need. The lower the age of the students, the greater the difficulty of classroom management [27]. It may also depend on how the desks are arranged, the number of pupils in the classroom, and the acoustics [28]. These factors also predispose teachers to incorrect postures that may act as a cause or comorbidity of vocal disorders.

In 1949, Feldenkrais hypothesized that posture can affect the voice and be influenced. The author described how the hyoid bone and the muscles, which branch off it and the skull, influence the position of the larynx [29]. Posture can be identified as the position the body assumes in space to maintain static and dynamic balance [30]. It is influenced by psycho-emotional, biochemical, neurophysiological, and biomechanical components. The imbalance of these structures leads to pharyngeal tissue and the laryngeal muscles changes. It shifts the larynx and affects the passage of air within it. It causes a change in vocal control and pitch [31]. This review aims to evaluate if there is a correlation between posture and MTD. It also aims to study how a postural intervention influences this disorder.

## 2. Materials and Methods

This scoping review was compiled following the Preferred Reporting Items for Systematic Reviews and Meta-Analyses (PRISMA) guidelines for scoping review [32]. The protocol was not registered in PROSPERO due to the topic, which is not among the arguments accepted by the database.

Eligibility criteria were created as suggested by PRISMA for population (P), intervention (I), comparison (C), outcomes (O), and study design (s). The population included was composed of adults (above 18 and below 65 years of age) and their profession had to be within a school (teachers or professors). The population had to present MTD. The intervention had to be based on the improvement of the participant’s posture. The comparison was between different populations (such as teachers with MTD and people without MTD) or within the population (before or after the intervention). The outcomes were related to the postural analysis and voice evaluation. Techniques considered were the Voice Handicap Index (VHI), laryngoscopy, and audiometry, or photograms for postural analysis. Articles included had to be written in English; other languages were excluded. The country of origin was not a limit for inclusion or exclusion. Articles included had to be original; reviews, meta-analyses, abstracts, conference papers, chapters, and books were excluded. We also excluded case reports and case series to better assess the articles’ quality. All articles had to be published in peer-reviewed journals.

### 2.1. Search Strategy

The search was performed in May 2023 and articles published between 2005 and 2023 were included. The electronic databases PubMed, Scopus, Web of Science, Springer, Science Direct, ResearchGate, and Cochrane Register were searched. The PRISMA checklist is included as Appendix A.

Based on the aim of the review, a literature search strategy was developed using medical subject headings (MeSH) for the words “posture”, “taught”, “muscle tensive dysphonia”, “voice”, “voice disorders”, “teacher”, “dysphonia”, and “MTD”. The keywords were matched with the Boolean operators “AND” and “OR”.

### 2.2. Data Collection and Extraction

Two operators (RF and LP) carried out the data extraction taking into account the above-mentioned inclusion criteria. After the search of the articles in the databases, the articles were screened against the eligibility criteria. Articles were saved in the collection, organization, sharing, and referencing site ‘Zotero’ Your Personal research assistant (Available at: https://www.zotero.org. Accessed on 18 December 2023). The free version of the automatic tools on Covidence (Available at: https://www.app.covidence.org. Accessed on 18 December 2023) was used, in which the above-mentioned inclusion and exclusion criteria were included. The first screening was performed by the title, followed by the screening of the abstract and the full text. The two operators were not blind to the authors or affiliations of the articles screened. In case of disagreement between the two investigators, the study coordinator (GM) was involved to have a final consideration. After the screening process, data on population, treatment, study type, outcomes, and main results were extracted from any part of the manuscript. The data are discussed narratively.

### 2.3. Quality Assessment

The quality assessment was conducted by one investigator (RF). The Grading of Recommendation, Assessment, Development, and Evaluation (GRADE) framework was used. GRADE is a tool that allows rating the quality of evidence; it is a useful instrument for reviews to demonstrate the good or poor quality of the included studies [33]. It is a transparent and structured process that through questions allows rating [33], in this case, of papers. GRADE presents evaluation criteria covering all aspects of a randomized controlled trial, clinical trial, and qualitative study [34]. It is a tool that allows great evaluative freedom as the questions corresponding to the evaluation criteria are selected by the author depending on the review being conducted [34]. The important aspect of a paper being able to state that it is using GRADE is that the questions refer to the PICO system (Population, Intervention, Comparator, and Outcome) used in evidence-based medicine [35].

## 3. Results

The search in the different databases led to an initial result of 396 records. Of these, 95 were duplicates and were automatically removed. After the screening process against the eligibility criteria, eight articles were included in the study. The reasons for the exclusion of articles are represented in Figure 1.

### 3.1. Description of Studies

The studies included were four randomized controlled trials, one controlled clinical trial, one prospective cohort study, and two cross-sectional studies.

The total number of subjects examined in the eight studies was 303. Only six out of eight articles specified the number of men and women included in the study. There were 196 females, 25 males and of 82 the sex was not specified. According to the data collected, more than 66% of the participants were females. The average age of the participants was between 18 and 60 years.

Of the articles reviewed, many are common outcomes, such as the VHI, used by five out of eight articles. They covered postural analysis by observation or photograms, used by six out of eight articles. They also studied laryngoscopy with its variants, used by four out of eight articles, and covered audiometry, used by five out of eight articles.

Not all of the articles were mainly about the correlation of posture and MTD. However, they all had outcomes that are strongly related to the topic. The approaches to treating MTD are different. Some studies attempted to improve symptoms, others the correlation or probability of presentation. More details on the studies, outcomes, and results obtained are summarized in Table 1.

Andriollo et al. (2021) [36] evaluated the effect of pompage techniques in improving posture and perceived disability from neck pain and vocal disorders (also MTD) in teachers. Pompage techniques are a variant of pump techniques and are part of myofascial release techniques. Three treatments per week were performed for a total of 24 sessions. The results showed that the pompage group improved their posture and cervical spine position about the head and shoulder. A reduction in the perception of disability was not present in the control group. They focused on the posterior musculature of the cervical spine and did not evaluate the anterior musculature. The front muscle was affected by the techniques used and therefore underwent a passive improvement.

Cardoso et al. (2021) [37] evaluated myofascial techniques’ effectiveness in improving posture, muscle tension, and voice quality in teachers with hyperfunctional dysphonia (a subcategory of MTD). An improvement was detected after the myofascial release techniques session. It improved the alignment of the cervical spine and head with the shoulders and pelvis. There was also an improvement in the parameters used in voice evaluation. There is no improvement in the diaphragm muscle. The muscles found to be most hypertonic were the cricothyroid, the thyrohyoid, and the suprahyoid muscles in general.

Cardoso et al. (2020) [39], in another study, evaluated the association between posture, muscle tension, and voice disorders, and indicated that there is no interest in the effectiveness of treatment. The results showed that there are no statistically significant differences in the position of the cervical spine between teachers with voice disorders and those without. The difference in the tension of the laryngeal and hyoid muscles, in particular of thyrohyoid, cricothyroid, and suprahyoid muscles, is accentuated in the group of teachers with voice disorders and the voice assessment parameters are altered in the same group.

The study by Colla et al. (2022) [40] emphasized that teachers are subjected to excessive speech use and incorrect posture due to working environments that are not ergonomically suitable. They checked if self-rated values in the areas of vocal health, musculoskeletal disorders, and emotional disorders were more or less other than the reference parameters. There are higher values concerning voice impairment, musculoskeletal pain, and emotional disorders than the reference parameters. In particular, it can be seen that the most common pain is in the neck, shoulders, and lumbar region. An increase in values for these pains is associated with an increase in the perception of depression. The authors also say that an excessive and incorrect functional demand on the larynx can lead to decompensation of the cervical and laryngeal muscles with a postural change that leads to an anterior protraction of the head and an alteration of the airflow during phonation.

The study by Faralli et al. (2017) [41] focused on speech therapy and aimed to evaluate the effect of speech rehabilitation in subjects with MTD. A speech rehabilitation session was carried out every 15 days for 2 months, and the patient performed exercises at home between sessions. The results showed that, even at the end of speech therapy, some postural abnormalities remain. But, there is a strong correlation between the improvement of speech parameters and posture. Thus, doing a postural assessment before rehabilitation can greatly increase therapy’s effects. In particular, they noticed an improvement in the position of the cervical spine, which was forward. Both the muscles in the anterior and posterior musculature were too tense.

The article by Kooijman et al. (2005) [42] aimed to assess the relationship between laryngeal muscle hypertonus and voice disorders, including MTD, in teachers who take time off work for these problems. The results showed a direct correlation between posture and the results of the VHI and DSI. They emphasized the cricothyroid, thyrohyoid, and geniohyoid muscles were particularly hypertonic. The most common posture was with the head forward.

The study by Marszalek et al. (2012) [38] evaluated the effectiveness of osteopathic treatment in correlation with speech therapy in work-related dysphonia including MTD. The patients were given 30 min of osteopathic treatment. It focused on myofascial release techniques. Then, they had 30 min of voice therapy. During the therapy, they were taught exercises to do at home. Analysis of posture and voice before and after the treatment showed an initial hypertone, mainly at the cricothyroid, thyrohyoid, geniohyoid, and sternocleidomastoid muscle. The hypertone decreased significantly after the treatment, leading to an improvement of all postural and voice parameters.

Lastly, the study by Rantala et al. (2018) [43] aimed to study and evaluate the postures most commonly assumed by teachers and assess the correlation between them and voice disorders including MTD. After the postural analysis, they were given postural correction exercises, and two of them were asked to undergo an eye examination. The study found that those who rotated their head and back several times during working hours or raised their arms beyond their shoulders several times had a greater impairment from voice disorders. The disorders decreased as their posture improved. This makes it possible to hypothesize that the posture and gestures that teachers assume daily negatively affect voice health. A results summary is proposed in Table 2.

### 3.2. Articles Evaluation

As mentioned above, the articles were evaluated through GRADE. Given that GRADE also evaluates the treatment, the two cross-sectional studies by Cardoso et al. (2020) [39] and Colla et al. (2022) [40] were excluded from the evaluation. They are just statistical studies and do not evaluate a therapeutic intervention. All studies included in the review are of medium to high quality. Table 3 summarizes the answers to these questions.

## 4. Discussion

This literature review aims to investigate the correlation between posture and MTD in teachers and the possible interventions. It detected an alteration of the head’s alignment with the cervical spine, which is in anterior protraction with hypertonus of the suprahyoid muscles, of the cricothyroid, and an overload of the cervical musculature. There is a greater involvement of the sternocleidomastoid muscle. An ante and internal rotation position of the shoulders with increased dorsal kyphosis was also found. Also, the review detected that few studies proposed an intervention. The authors found that their intervention was feasible and effective.

Other studies found similar results. Lawell et al. (2012) [44] found, in a group composed of no teachers, a higher position of the larynx and hyoid bone than non-sufferers from MTD, due to the excessive contraction of the supra-hyoid muscles involved in the phonation process and MTD. This partially explains the predisposition of women to MTD, as women tend to have a higher larynx than men. They also have a breathing pattern that favors the accessory muscles. Women also have thinner vocal folds that predispose them to vocal tract disorders including dysphonia [28].

The change in craniocervical position is found in singers when they move from a low to a high note and vice versa. It is because the anatomical relationships between the cervical spine, the larynx, the hyoid, and the sternum alter the air amount and pressure needed for the required intonation [15]. As the cervical spine reduces its lordosis and the head is protracted forward, the voice will become higher pitched. But, it will also become more expensive to sustain and worse in quality. This leads to hyperfunctionality of the area and then to dysphonia. In the case of teachers, a study conducted by Cardoso et al. (2021) [37] found no difference between the craniocervical position of MTD sufferers and asymptomatic ones. These results are explained by Rantala et al. (2018) [43], who examined the postures most commonly assumed by teachers at work. The result is an anterior and superior closure of the shoulders, an internal rotation of the shoulders, and an anterior protraction of the head with an increase in dorsal kyphosis and lumbar lordosis. Colla et al. (2022) [40] analyzed the results of questionnaires on quality of life, job satisfaction, and vocal ability. They looked at subjects with vocal problems and a healthy larynx and also found this same posture [45]. The results showed many teachers had a negative perception of their quality of life at work related to very high stress and anxiety, with a postural and vocal modification and a shift in emotion.

A very similar postural posture was found in asthmatic patients. The study by Almeida et al. (2013) [45] shows that asthma sufferers tend to have a more pronounced dorsal kyphosis with an ante position of the head and an internal rotation of the shoulders. This posture follows a reduction in the function of the diaphragm. It is due to weak abdominal muscles with overactivation of the anterior cervical muscles and an overloading of the posterior ones. This lowers airflow and vocal emission quality, and predisposes subjects to voice disorders [45].

Studies show a correlation between posture and MTD. It is demonstrated through questionnaires, video laryngoscopy, and postural assessment. These methods reveal improved vocal fold mechanics. They also show that treatments to improve posture lead to better voice quality. The correlation between craniocervical misalignment and MTD is very useful for preventing and improving other musculoskeletal disorders [46]. Their study showed on X-ray that patients with a straightened, forward-protruding neck had a greater reduction in inter-somatic spaces than subjects with a normal neck.

The treatments are mostly characterized by manual techniques combined with vocal exercises. For example, the study by Marszalek et al. (2012) [38] analyzed the effectiveness of osteopathic myofascial techniques in improving vocal disorders. They found it by using VHI and video stroboscopy. At the end of the planned sessions, a normalization of the craniocervical relationship was found, as an increase in the symmetry of the thyroid muscles, and a relaxation of the cricothyroid. It was related to an improvement in vocal parameters found by the patients with the VHI. In a study not included in this review, the authors improved the quality of the voice indirectly [47]. They did it with a treatment for balance disorders that improved posture [47]. This highlights the importance of a postural treatment to act, indirectly, to the voice. A recent review suggests that respiratory exercises could improve vocal function, and can easily be performed with a postural treatment [48].

This review has its limitations. The number of articles included is small and of different types, with relative difficulty in qualitative and comparative evaluation. Consequently, a meta-analysis was not performed. Furthermore, the articles on the topic are not recent, and more studies are necessary to better support the findings. The clinical trial does not feature randomization of subjects with a consequent reduction in the quality of the study. The sample evaluated by some studies is small, and in some of them MTD is included but not the only one among the voice disorders analyzed. The biological sex most present is female, reducing the ability to predict the results in the male sex. The assessment methods were valid and similar between the articles, but the articles did not all focus on the posture–MTD correlation. This may have affected the specificity of the assessment of the correlation.

## 5. Conclusions

A correlation exists between posture and MTD in teachers. This review highlights a general alteration in the alignment of the head with the cervical spine and an ante and internal rotation of the shoulders with increased dorsal kyphosis. Pompage and myofascial techniques, improving posture, could be useful in reducing MTD symptoms. The results suggest that it is necessary to continue studying this correlation. We need more recent, valid, and precise results.

## Figures and Tables

**Figure 1 jfmk-09-00060-f001:**
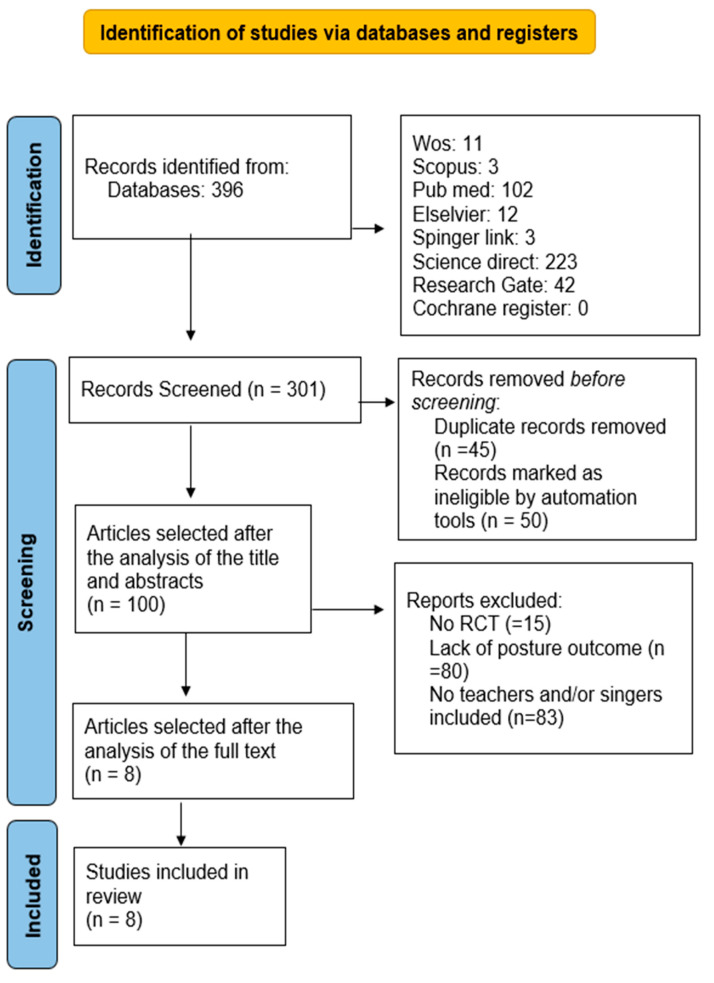
Flow chart. Note: RCT = randomized controlled trials.

**Table 1 jfmk-09-00060-t001:** Collection of articles selected for review.

First Author, Year	Patient (Number), (f), [m]; Age; Years of Employment	Aims	Outcomes	Treatment	Results
Randomized Controlled Trials					
Andriollo (2021) [36]	(56) 28 controls 21–60 years 5–18 years, aver. 9.1	Study the changes after pompage in teachers with vocal complaints	PPT, audiometry, laryngoscopy, NDI, visual postural analysis, craniocervical frames	No	Improvement in NDI, craniocervical posture, and laryngoscopy.
Cardoso (2021) [37]	(18) [6] 12 controls 27–60 years No info	Study the effect of myofascial release on teachers’ posture and MTD	Frames, palpation of muscle tension, algometer, voice aerodynamic, acoustic-auditory perceptual voice	Myofascial release	Improvement in vertical alignment of the head, pelvis, scapula, and vocal parameters.
Marszalek (2012) [38]	(38) [2] 38–59 years No info	Assess the use of osteopathic procedures in the diagnosis and treatment of MTD	Laryngostroboscopy MPT calculation, VHI visual postural assessment performed by an osteopath	Osteopathic treatment and speech therapy	Improvement in head and neck position correlated with more improvement in VHI than in MPT. Hypertone at the sternocleidomastoid level of the geniohyoid and cricothyroid.
Cross-Sectional					
Cardoso (2020) [39]	(18) [6] 11 controls 27–60 years No info	Study the differences in posture, muscle tension, and voice	Postural assessment (photograms, palpation), aerodynamic, acoustic, and perceptual assessment of voice	No treatment	Teachers with voice disorders had greater hypertone of laryngeal and hyoid muscles.
Colla (2022) [40]	57 21–60 years No info	Study the results of self-assessment of teachers with and without voice disorders	Videolaryngoscopic evaluation, Hearing screening, VAPP, V-RQOL, VHI, VoiSS, VTDS, HADS, URICA-VOICE, NMQ	Self-assessment results, reference parameters	There are higher values concerning voice impairment, musculoskeletal pain, and emotional disorders than the reference parameters.
Prospective Cohort					
Faralli (2017) [41]	(34) [3] 15–35 years	Examine postural control in MTD after a treatment based on speech rehabilitation	Romberg, laryngoscopy, laryngostroboscopy, VHI, acoustic voice analysis through CSLM and MDVP, postural analysis through SPeV, ECR, and ECO	Rehabilitation speech	Altered proprioceptive signals at the cervical level altered posture and voice, which in turn was altered by them. Speech rehabilitation led to improvement in subjective symptoms.
Qualitative					
Kooijman (2005) [42]	25 No info	Study the relationship between body posture and voice disorders in teachers	VHI and DSI, Postural assessment, palpation of muscle tension	No treatment	Some postures predispose to voice disorders: posteriorly shifted weight, anterior protraction of the head, hypertonia of the sternocleidomastoid, and cricothyroid.
Controlled Clinical Trials					
Rantala (2018) [43]	(32) [8] 27–57 years Up to 10 years	Evaluate the associations between the voice and working postures of teachers	VEAW, VHI, and evaluation of the voice through recording it during a reading and one-day class	No treatment, posture correction	Teachers with greater vocal disorders tended to tilt their heads, rotate their torsos, and raise their arms when explaining the lesson.

Note: VHI = Voice Handicap Index, DSI = Dysphonia Severity Index, CSLM = Computerized Speech Laboratory Model, MDVP = Multi-Dimensional Voice Program software, MPT = maximum phonation time, PPT = pressure pain threshold, NDI = neck disability index, VEAW = whole voice ergonomic risk assessment, VAPP = vocal activity and participation profile, V-RQOL = voice-related quality of life, VoiSS = voice symptom scale, VTDS = vocal tract discomfort scale, HADS = hospital anxiety and depression scale, URICA-VOICE = University of Rhode Island, NMQ = Nordic musculoskeletal questionnaire.

**Table 2 jfmk-09-00060-t002:** Summary of the results of the included studies.

	Andriollo (2021) [36]	Cardoso (2021) [37]	Marszalek (2012) [38]	Cardoso (2020) [39]	Colla (2022) [40]	Faralli (2017) [41]	Kooijman (2005) [42]	Rantala (2018) [43]
OUTCOME: VHI								
Mean value pre-intervention/treatment or baseline evaluation	X	X	Total: undeclared	X	Total: 22.07 ± 17.12	Total: 42.77 ± 8.71	Total: 72	Physical subsection: 13.4 ± 5.72
Mean value post-intervention/treatment	X	X	Total: lower of 19.6	X	No intervention	Total: 26.72 ± 8.11	No intervention	Physical subsection: 7.9 ± 5.32
OUTCOME: Postural analysis without palpation								
Mean value pre-intervention/treatment	Evaluation with SAPO [37] HFE: 158.02 AP: 35.31	Evaluation with SAPO [37] HAH: 2.29 HAASIS: 1.67 HAA: 1.66 HAP: −2.75 HAST3: −1.59 VAB: 1.25 VAHA: 2.54	X	Evaluation with SAPO [37]. VCG: HAH: 1.78 ± 1.37 HAASIS:1.42 ± 1.13 HAA: 1.55 ± 1.28 HAP: −3.21 ± 5.50 HAST3:−1.34 ± 4.18 VAB: 1.20 ± 1.03 VAHA: 2.68 ± 2.15	X	Postural analysis through SPeV [41] SEC: 410.73 SECR: 544.69	X	X
Mean value post-intervention/treatment	HFE: 155.33 AP: 35.28	HAH: 1.56 HAASIS: 1.63 HAA: 1.59 HAP: −2.66 HAST3: −1.60 VAB: 1.22 VAHA: 2.14	X	WVCG: HAH: 1.17 ± 2.76 HAASIS:0.81 ± 2.30 HAA: 0.87 ± 2.30 HAP: −3.78 ± 4.26 HAST3: −1.83 ± 3.89 VAB: 0.45 ± 2.01 VAHA: 3.23 ± 3.74	X	SEC: 333.15 SECR: 465.07	X	X

Note: SAPO = Postural Evaluation Software [37], SVeP = standard vestibology platform [41], HFE = head flexion-extension, AP = anteriorized posture, VCG = voice complain group, WVCG = without voice complain group, HAA = horizontal alignment of acromion, HAASIS = horizontal alignment of anterior superior iliac spine, HAH = horizontal alignment of the head, HAP = horizontal alignment of the pelvis, HAST3 = Horizontal asymmetry of the scapula about T3, VAB = vertical alignment of the body, VAHA = vertical alignment of the head with the acromion [37], SEC = sway area with eyes closed, SECR = sway area head retroflex test with eyes closed [41]; X = no information.

**Table 3 jfmk-09-00060-t003:** Article evaluation summary.

Question	Andriollo, 2021 [36]	Cardoso, 2021 [37]	Faralli, 2017 [41]	Kooijman, 2005 [42]	Marszalek, 2012 [38]	Rantala, 2018 [43]
Are the characteristics of the patients in terms of gender, age, and average weight specified?	YES, not gender	YES, not gender	YES	YES, not gender	YES	YES
Is there a control group present?	YES	YES	NN	NN	NO	High
Are the operators and measurers blind?	YES	YES	NO	NN	NO	Mod
Are the descriptions of the treatments or any protocols done on the patients explained in a clear and illustrative manner?	YES	YES	YES	YES	YES	Mod
Are all outcomes listed and explained?	YES	YES	NO	YES	YES	High
How reproducible is it considering patient characteristics, outcomes, and interventions?	High	High	Mod	Mod	High	High
How high is the risk of bias considering the validity of the selected outcomes, the blindness of the operators, and the explanation of the interventions?	Low	Low	Low	Mod	Mod	Mod
Quality	High	High	Mod	Mod	Mod	Mod

Note: Mod = moderate, NN = not necessary.

## Data Availability

All data extracted are included in the tables and figures.

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
