# Peer review of "The Relationship between Posture and Muscle Tensive Dysphonia in Teachers: A Systematic Scoping Review"

_jfmk, 2024, doi:10.3390/jfmk9020060_

Round 1
Reviewer 1 Report
Comments and Suggestions for Authors
The relationship between posture and muscle tensive dysphonia in teachers: a scoping review
Very interesting, actual and innovative work with a lot of raising questions.
1. Introduction
Excellent written text. It has everything necessary to acquaint the reader with the background of the study and defines its tasks.
2. Materials and Methods
This section was compiled following the preferred scheme. The literature search strategy was developed using medical subject headings (MeSH) for the used important words.
It is clearly described the subsections 2.1 Data collection and extraction and 2.2 Quality assessment.
3. Results
The data collection scheme is described in detail with excellent tabular visualization and a clear block diagram. Logically arranged subsections with adequate content.
4. Discussion
Sounds good and accepted. This review has its limitations which are described here.
5. Conclusions
The authors write (336 - 338): “Highlighting and finding this correlation is important to develop and increase the possibilities of prevention and treatment through manual and speech therapies that can improve the quality of daily and working life of those affected.”
That sounds unacceptably general. It is necessary to specify these "possibilities of prevention and treatment". This should be done with a clear description of these possibilities.
6. The Abstract and Conclusions must be improved.
Improvement. In the abstract and conclusions must be underline clearly the new results and conclusions presented from the authors which differ from those obtained till now. It must be underline the main authors contributions. In the Abstract must be included the main conclusions and the authors must underline the own approach contributions and the basic benefits from presented results in practice.
7. If this is possible the Authors can add more recent references – 9 of 38 references are from the last five years.
I hope that the proposed corrections will increase the quality of the manuscript and possibly its citability.
Author Response
Reviewer 1
Very interesting, actual and innovative work with a lot of raising questions.
Reply: thank you for the comment, thank you also for the time spent on the manuscript.
- Introduction
Excellent written text. It has everything necessary to acquaint the reader with the background of the study and defines its tasks.
Reply: thank you very much for the comment.
- Materials and Methods
This section was compiled following the preferred scheme. The literature search strategy was developed using medical subject headings (MeSH) for the used important words.
It is clearly described the subsections 2.1 Data collection and extraction and 2.2 Quality assessment.
Reply: thank you very much for the comment.
- Results
The data collection scheme is described in detail with excellent tabular visualization and a clear block diagram. Logically arranged subsections with adequate content.
Reply: thank you very much for the comment.
- Discussion
Sounds good and accepted. This review has its limitations
which are described here.
Reply: thank you very much for the comment.
- Conclusions
The authors write (336 - 338): “Highlighting and finding this correlation is important to develop and increase the possibilities of prevention and treatment through manual and speech therapies that can improve the quality of daily and working life of those affected.”
That sounds unacceptably general. It is necessary to specify these "possibilities of prevention and treatment". This should be done with a clear description of these possibilities.
Reply: thank you for this comment. We agree with the Reviewer, the sentence was too general and a deeper analysis, according to us, was not appropriate for this section of the manuscript. The conclusion has been written in a more concise way. We hope the Reviewer agree with our decision.
- The Abstract and Conclusions must be improved.
Improvement. In the abstract and conclusions must be underline clearly the new results and conclusions presented from the authors which differ from those obtained till now. It must be underline the main authors contributions. In the Abstract must be included the main conclusions and the authors must underline the own approach contributions and the basic benefits from presented results in practice.
Reply: thank you for this comment. We modified the two sections according the Reviewer comments. We hope that now they result improved.
- If this is possible the Authors can add more recent references – 9 of 38 references are from the last five years.
Reply: thank you for this comment. Unfortunately the number of manuscripts on this topic is limited. We highlighted this point in the limits of the study.
I hope that the proposed corrections will increase the quality of the manuscript and possibly its citability.
Reply: Thank you, the Reviewer for its time and the comments provided.
Reviewer 2 Report
Comments and Suggestions for Authors
We, as authors, congratulate you on the article. Here are some comments to try to improve it. Minor details:
- The introduction is correct.
- The methodology is brief. I see it necessary to include the definition used for MTD, and the minimal required assessment (instrumental, videostroboscopic, ENT, rehabilitative, PRAAT, physical examination...?).
- Was the PICO format not used?
- Figure 1 shows many more sources than those cited in the methodology.
- Include in supplementary material: PRISMA checklist. Summary table of the search methodology used. Define each search key with the words used and the results obtained in each of them.
- A start date for the search was not used: from ... to May 2023? This could influence patient selection and MTD diagnostic criteria.
- Line 156, change of verb tense. Past tense should be used.
- Review the verb tense throughout point 3.1
- I miss data in the summary tables of the studies. Mean age, control group, etc...
- The discussion is adequate.
Comments on the Quality of English Language
OK
Author Response
Reviewer 2.
We, as authors, congratulate you on the article. Here are some comments to try to improve it. Minor details:
Reply: Thank you very much for the comment and for the time spent in this manuscript.
- The introduction is correct.
Reply: thank you for the comment
- The methodology is brief. I see it necessary to include the definition used for MTD, and the minimal required assessment (instrumental, videostroboscopic, ENT, rehabilitative, PRAAT, physical examination...?).
Reply: thank you for the suggestion. We importantly improved the methodology, we hope the reviewer appreciate the work done.
- Was the PICO format not used?
Reply: thank you for the comment. We added detailed information on the population, intervention, comparison and outcomes.
- Figure 1 shows many more sources than those cited in the methodology.
Reply: thank you for this comment. We wrote the manuscript the protocol before the articles collection. After this first phase, we noted a poor number of manuscripts on this topic, consequently we screened other databases. We changed our method and added all the databases searched.
- Include in supplementary material: PRISMA checklist. Summary table of the search methodology used. Define each search key with the words used and the results obtained in each of them.
Reply: thank you for the suggestion. We added PRISMA checkist in the supplementary material.
- A start date for the search was not used: from ... to May 2023? This could influence patient selection and MTD diagnostic criteria.
Reply: We decided to start to collect articles from 2005 when the methodology started to be more consistent. It is now specified in the study. We hope the Reviewer understand our decision
- Line 156, change of verb tense. Past tense should be used.
- Review the verb tense throughout point 3.1
Reply: thank you for the point. We corrected it.
- I miss data in the summary tables of the studies. Mean age, control group, etc...
Reply: thank you for the suggestion. We added some information in the table, we hope the Reviewer appreciate the work done.
- The discussion is adequate.
Reply: thank you for the point.
Reviewer 3 Report
Comments and Suggestions for Authors
In the work was raised an interesting problem relationships between posture and muscle tensive dysphonia in teachers. Work complied with the principles for systematic reviews (PRISMA). Based on the presented systematic review, recommendations for use in clinical practice have been presented. I recommend publishing the work in its current form.
Comments on the Quality of English Language
In the work was raised an interesting problem relationships between posture and muscle tensive dysphonia in teachers. Work complied with the principles for systematic reviews (PRISMA). Based on the presented systematic review, recommendations for use in clinical practice have been presented. I recommend publishing the work in its current form.
Author Response
In the work was raised an interesting problem relationships between posture and muscle tensive dysphonia in teachers. Work complied with the principles for systematic reviews (PRISMA). Based on the presented systematic review, recommendations for use in clinical practice have been presented. I recommend publishing the work in its current form.
Reply: thank you for the comment and the time spent on the manuscript.
Reviewer 4 Report
Comments and Suggestions for Authors
Review jfmk-2876446-peer-review-v1
The paper The Relationship between Posture and Muscle Tensive Dysphonia in Teachers: A Systematic Scoping Review is interesting. In this scoping review, researchers examine existing literature to understand the extent of knowledge on how posture influences muscle tension dysphonia in teachers. The findings may shed light on potential interventions or preventive measures to mitigate the condition in this specific population.
To the paper I have minor comments, which should contribute to a better understanding and readability of the work.
Material and Methods: Why are there databases listed in lines 107-108 that differ from those in Figure 1? Additionally, why didn't the authors search the EBSCO database?
In section 2.1, I suggest writing the initials of the authors who performed each component of the review.
In section 2.2, please provide a description of GRADE along with references to the literature. Additionally, could you specify who conducted the quality assessment?
Results. Shouldn't the captions be placed under the figures? In Figure 1, what does 'RCT' refer to? Additionally, why is there mention of singers if this element was not discussed in the materials and methods section?
In my opinion, Table 1 should be placed at line 165. Additionally, considering that only 8 papers were included in the analysis, Table 1 should provide more detailed information. Could you clarify what 'RCT' stands for? Furthermore, I believe it would be beneficial to include explanations for each type of study. Regarding the third column, it would be useful to include information on the gender distribution, duration of employment, and specific issues encountered by the participants, such as their characteristics. I also suggest grouping the papers according to the study type key, then the table will have the form and add aim of the paper.
|
Study |
Patient |
Aim |
Outcomes |
Treatment |
Results |
|
RCT |
|||||
|
Marszałek (2012) |
|
|
|
|
|
|
|
|
|
|
|
|
|
Cross-sectional |
|||||
|
|
|
|
|
|
|
In my opinion, it would be good to see some numerical values in the results.
Table 2 – why in the last column in the third and 6th row is the word Hig h, what does it mean?
Author Response
The paper The Relationship between Posture and Muscle Tensive Dysphonia in Teachers: A Systematic Scoping Review is interesting. In this scoping review, researchers examine existing literature to understand the extent of knowledge on how posture influences muscle tension dysphonia in teachers. The findings may shed light on potential interventions or preventive measures to mitigate the condition in this specific population.
To the paper I have minor comments, which should contribute to a better understanding and readability of the work.
Reply: thank you very much for the comments and the time spent on the manuscript.
Material and Methods: Why are there databases listed in lines 107-108 that differ from those in Figure 1? Additionally, why didn't the authors search the EBSCO database?
Reply: thank you for this comment. We wrote the manuscript the protocol before the articles collection. After this first phase, we noted a poor number of manuscripts on this topic, consequently we screened other databases. We changed our method and added all the databases searched.
About the question on EBSCO database, we have problem to access in that database. Anyway, thank you for the suggestion.
In section 2.1, I suggest writing the initials of the authors who performed each component of the review.
Reply: thank you for the comment. We added the initials of the two investigators that worked in this stage.
In section 2.2, please provide a description of GRADE along with references to the literature. Additionally, could you specify who conducted the quality assessment?
Reply: we added who conducted the quality assessment.
-Results. Shouldn't the captions be placed under the figures? In Figure 1, what does 'RCT' refer to? Additionally, why is there mention of singers if this element was not discussed in the materials and methods section?
Reply: thank you for the suggestion. We moved the title under the figure and added the notes for RCT.
-In my opinion, Table 1 should be placed at line 165. Additionally, considering that only 8 papers were included in the analysis, Table 1 should provide more detailed information. Could you clarify what 'RCT' stands for? Furthermore, I believe it would be beneficial to include explanations for each type of study. Regarding the third column, it would be useful to include information on the gender distribution, duration of employment, and specific issues encountered by the participants, such as their characteristics. I also suggest grouping the papers according to the study type key, then the table will have the form and add aim of the paper.
Reply: thank you for the suggestion, we modified the table according the Reviwer indications. Thank you.
|
Study |
Patient |
Aim |
Outcomes |
Treatment |
Results |
|
RCT |
|||||
|
Marszałek (2012) |
|
|
|
|
|
|
|
|
|
|
|
|
|
Cross-sectional |
|||||
|
|
|
|
|
|
|
In my opinion, it would be good to see some numerical values in the results.
Reply: we tried to extrapolate some data, they are in table 3.
Table 2 – why in the last column in the third and 6th row is the word Hig h, what does it mean?
Reply: thank you for the comment. We corrected it in high.